# Investigating Urine Biomarkers in Detrusor Underactivity and Detrusor Overactivity with Detrusor Underactivity Patients

**DOI:** 10.3390/biomedicines11041191

**Published:** 2023-04-17

**Authors:** Yuan-Hong Jiang, Jia-Fong Jhang, Ya-Hui Wu, Hann-Chorng Kuo

**Affiliations:** 1Department of Urology, Hualien Tzu Chi Hospital, Buddhist Tzu Chi Medical Foundation, Hualien 970, Taiwan; 2Department of Urology, School of Medicine, Tzu Chi University, Hualien 970, Taiwan

**Keywords:** detrusor underactivity, detrusor overactivity with detrusor underactivity, urine biomarker, oxidative stress

## Abstract

Bladder inflammation and tissue hypoxia were considered important pathognomonic bladder features in detrusor underactivity (DU) and detrusor overactivity (DO) patients. This study investigated urine inflammatory and oxidative stress biomarker levels in DU and DO with DU (DO-DU) patients. Urine samples were collected from 50 DU and 18 DO-DU patients, as well as 20 controls. The targeted analytes included three oxidative stress biomarkers (8-OHdG, 8-isoprostane, and total antioxidant capacity [TAC]) and 33 cytokines. DU and DO-DU patients had different urine biomarker profiles from controls, including 8-OHdG, PGE2, EGF, TNFα, IL-1β, IL-5, IL-6, IL-8, IL-10, IL-17A, and CXCL10. Controlling for age and sex, multivariate logistic-regression models revealed that 8-OHdG, PGE2, EGF, IL-5, IL-8, IL-10, and TAC were significant biomarkers for diagnosing DU. In DU patients, urine TAC and PGE_2_ levels were positively correlated with detrusor voiding pressure. In DO-DU patients, urine 8-OHdG, PGE_2_, IL-6, IL-10, and MIP-1α levels were positively correlated with maximal urinary flow rate, while urine IL-5, IL-10, and MIP-1α were negatively correlated with the first sensation of bladder filling. Urine inflammatory and oxidative stress biomarker analysis provides a non-invasive and convenient approach for important clinical information in DU and DO-DU patients.

## 1. Introduction

Detrusor underactivity (DU) is a common but under-researched cause of lower urinary tract symptoms (LUTS). It possibly accounts for 25–48% and 12–24% of these symptoms in elderly men and women, respectively [1]. DU is a urodynamic diagnosis defined by the International Continence Society as “low detrusor pressure or short detrusor contraction time, usually in combination with a low urine flow rate resulting in prolonged bladder emptying and/or a failure to achieve complete bladder emptying within a normal time span [2]”. Both the imprecise operational definition and indefinite diagnostic criteria impede advanced research on DU [3].

The mechanisms involved in DU include myogenic failure, efferent nerve dysfunction, afferent nerve dysfunction, and brain/spinal cord dysfunction [4,5]. Multiple etiological factors are likely in DU, including nervous system injuries or diseases, diabetes mellitus, bladder outlet obstruction (BOO), aging, and other miscellaneous factors [4]. A study investigating bladder mucosa biopsies from DU patients found that urothelial dysfunction, increased suburothelial inflammation, and altered sensory protein expressions were the prominent pathological features [6]. Recent systemic review [3] and clinical research [7] both highlighted the potential roles of hypoxia and ischemia in causing bladder damage and inflammation and, ultimately, DU. One hypothesis is that long-term bladder ischemia originating from overactive bladder (OAB)/detrusor overactivity (DO) might make the disease progress to detrusor hyperactivity with impaired contractility (currently superseded by the term of DO with DU (DO-DU) [2]), and ultimately DU [8]. Bladder inflammation, tissue hypoxia, and oxidative stress are thought to be important pathologic features of DU [7,9]. However, there were few studies investigating the related urine biomarkers in DU patients.

Biomarkers expressed in urine specimens are attractive because of their non-invasiveness and their potential roles in revealing information inside the bladder. Increased urine inflammatory biomarkers [10], nerve growth factor, brain-derived neurotrophic factor (BDNF), and adenosine triphosphate [11] were reported in OAB patients. This might help in their diagnosis and treatment. Chen et al. first disclosed the potential prognostic roles of urine biomarkers in DU patients [12]. Urine prostaglandin E_2_ (PGE_2_) and BDNF levels in DU patients with bladder function recovery were significantly higher than those in the control group. However, we lacked a comprehensive investigation of urine biomarkers in DU and DO-DU patients. Additionally, the diagnostic roles of urine biomarkers have not been established.

Bladder inflammation and tissue hypoxia were important pathognomonic bladder features in DU patients, and there is a hypothetical transition from DO-DU to DU. Therefore, this study investigated urine inflammatory and oxidative stress biomarkers levels in DU and DO-DU patients.

## 2. Materials and Methods

### 2.1. Patients and Investigation of the Clinical Characteristics

From October 2017 to March 2021, we prospectively enrolled 50 DU patients and 18 DO-DU patients at the Department of Urology at a single medical center. All patients received VUDS, which determined the clinical diagnosis. The diagnostic details of VUDS were interpreted according to the terminology reported by the International Continence Society [2,13]. The study recorded various parameters of VUDS, including cystometric bladder capacity (CBC), bladder compliance, first sensation of bladder filling (FSF), fullness sensation (FS), detrusor voiding pressure (Pdet), maximal urinary flow rate (Qmax), corrected maximal urinary flow rate (cQmax, calculated as the product of Qmax and the square root of CBC), voided volume (Vol), post-void residual volume (PVR), and voiding efficacy (VE, defined as Vol divided by CBC).

The International Continence Society defined DU as “low detrusor pressure or short detrusor contraction time, usually in combination with a low urine flow rate resulting in prolonged bladder emptying and/or a failure to achieve complete bladder emptying within a normal time span”. DO-DU was defined as “urodynamic DO on filling cystometry in combination with urodynamic DU on pressure-flow studies [2]”. In this study, we set stricter conditions to enroll more typical study patients. The diagnostic criteria for DU were composed of a low Pdet, a low Qmax, a large PVR (>300 mL), and a low VE (<33%) without significant evidence of anatomical BOO in cystoscopy. The diagnostic criteria for DO-DU were composed of the presence of urodynamic DO during the storage phase, a low Pdet, a low Qmax, a large PVR (>100 mL), and a low VE (<50%) without significant evidence of anatomical BOO in cystoscopy. Enrolled study patients had the corresponding lower urinary tract diseases for at least three months. The International Prostate Symptom Score (IPSS), comprising total IPSS, IPSS storage subscore (IPSS-S), and IPSS voiding subscore (IPSS-V), was administered to all patients. In addition, a control group of 20 women with stress urinary incontinence, who did not exhibit substantial LUTS (IPSS ≤ 6) or significant storage or voiding dysfunction on VUDS, were also included in the study.

We excluded patients who need a long-term indwelling urethral catheter but not patients with clean intermittent catheterization. The other exclusion criteria included active symptomatic urinary tract infection, interstitial cystitis, a recent history of bladder surgery or traumatic injury, a history of urinary tract malignancy or tuberculosis, a history of nephrotic or nephritic syndrome, urolithiasis, and/or impaired renal function (serum creatinine > 2.0 mg/dL).

### 2.2. Assessment of Urine Biomarker Levels

Urine samples were collected from all enrolled study patients and controls. Urine was self-voided when the subjects reported a full bladder sensation. Single urethral catheterization to collect urine samples was conducted for patients who were unable to void. To ensure an infection-free status, a urinalysis was conducted before storing the fresh urine samples. A volume of 50 mL of urine was immediately placed on ice and transported to the laboratory for processing. After centrifugation at 1800 rpm for 10 min at 4 °C, the resulting supernatants were divided into 1.5-mL aliquots (1 mL per tube) and stored at a temperature of −80 °C. For subsequent measurements, the frozen urine samples were centrifuged at 12,000 rpm for 20 min at 4 °C, and the resulting supernatants were utilized.

#### 2.2.1. Quantification of Inflammatory Cytokines

The laboratory procedures used to investigate inflammatory cytokines, chemokines, and neurotrophins in the urine samples were similar to those described in our previous study [14]. The levels of the targeted analytes in the urine samples were measured using a Milliplex^®^ Human cytokine/chemokine magnetic bead-based panel kit (Millipore, Darmstadt, Germany). The study measured a total of 32 analytes of interest. EGF, eotaxin, G-CSF, GM-CSF, IFNα2, IFNγ, CXCL10, MCP-1, MIP-1α, MIP-1β, RANTES, IL-1RA, 1L-1α, IL-1β, IL-2, IL-3, IL-4, IL-5, IL-6, IL-7, IL-8, IL-10, IL-12p40, IL-12p70, IL-13, IL-15, IL-17A, TNFα, TNFβ, and VEGF were measured using the multiplex kit with the catalog number HCYTMAG-60K-PX30. A nerve growth factor was measured with the multiplex kit catalog number HADK2MAG-61K, and BDNF was measured with the multiplex kit catalog number HNDG3MAG-36K. The following laboratory procedures for the quantification of these targeted analytes were performed similarly to those in our previous study [14,15].

#### 2.2.2. Quantification of PGE2

Urine PGE_2_ level was measured using a high-sensitivity ELISA kit (Cayman, MI, USA), according to the manufacturer’s instructions. The detailed procedures were similar to those reported in a previous study [12].

#### 2.2.3. Quantification of Oxidative Stress Biomarkers

The study measured three oxidative stress biomarkers in urine samples, including 8-hydroxy-2-deoxyguanosine (8-OHdG), 8-isoprostane, and total antioxidant capacity (TAC), following the instructions provided by the respective manufacturers (8-OHdG ELISA kit from Biovision, Waltham, MA, USA; 8-isoprostane ELISA kit from Enzo, Farmingdale, NY, USA; and Total Antioxidant Capacity Assay Kit from Abcam, Cambridge, MA, USA). The laboratory procedures used for these measurements were similar to those described in a previous study [16].

### 2.3. Statistical Analysis

Continuous variables were presented as means along with their standard deviations, and categorical variables were presented as percentages and numbers. Values outside the range of mean ± three standard deviations were considered outliers and were excluded from the analysis of each biomarker in either the study or control group [14,15]. ANOVA was utilized to analyze the differences in clinical data and the levels of biomarkers in urine across groups, followed by post-hoc analysis. Biomarkers with mean values in study groups below the minimum detectable concentrations, as specified by the assay manufacturer, were also excluded for subsequent analysis. Post-hoc power calculation was performed in the biomarker with significant differences between the study and control groups. To assess the correlation between clinical characteristics and urine biomarker levels, a linear regression analysis using the Pearson correlation was conducted. Multivariate logistic regression models were created to control for confounding factors for each biomarker, and odds ratios were calculated. The calculations were performed using SPSS Statistics for Windows, Version 20.0 (IBM Corp., Armonk, NY, USA). Statistical significance was set at a *p*-value less than 0.05.

## 3. Results

Table 1 shows the characteristics and VUDS parameters of eligible study and control participants. A significant difference in gender was noted among the groups. DU and DO-DU patients, both with similar ages to the controls, had significantly higher IPSS than controls (21.1 ± 6.6, and 20.2 ± 10.4 vs. 3.2 ± 1.5, both *p* < 0.001). According to the corresponding VUDS diagnostic criteria set in this study, all three groups of patients had distinct VUDS presentations.

Table 2 shows the targeted urine biomarker levels among DU, DO-DU patients, and controls. For each targeted analyte, the numbers of outliers within groups ranged from 0 to 5 in DU patients, 0 to 3 in DO-DU patients, and 0 to 1 in controls. The urine biomarker profiles of DU and DO-DU patients were different from those of controls, including 8-OHdG, PGE2, EGF, TNFα, IL-1β, IL-5, IL-6, IL-8, IL-10, IL-17A, and CXCL10. Post-hoc power analysis reported 90.7%, 99.8%, 61.5%, 89.4%, 87.8%, 79.6%, 78.8%, 96.5%, 92.5%, 76.6%, and 85.5% power (with an alpha value of 0.05) in the evaluation of 8-OHdG, PGE2, EGF, TNFα, IL-1β, IL-5, IL-6, IL-8, IL-10, IL-17A, and CXCL10 levels in DU patients, respectively. Figure 1 demonstrates significant analytes discriminating DU, DO-DU patients, and controls. There was no significant difference in urine biomarker profiles between DU patients with different etiologies (neurogenic or non-neurogenic).

By adjusting for age and sex, multivariate logistic-regression models were used to reveal the odds ratio (OR) of targeted analytes (Table 3). Urine biomarkers that significantly distinguished DU patients from controls included PGE_2_ (OR = 2.165), 8-OHdG (OR = 1.557), IL-5 (OR = 1.544), IL-10 (OR = 1.318), IL-8 (OR = 1.291), TAC (OR = 0.904), and EGF (OR = 0.826) (all *p* values < 0.05).

Table 4 and Table 5 demonstrate the correlation coefficients between urine biomarker levels and VUDS parameters in DU patients and DO-DU patients, respectively. In DU patients, the significant correlations were weak; however, in DO-DU patients, the significant correlations were moderate to strong. In DU patients, urine TAC and PGE_2_ levels were positively correlated with Pdet, and the urine IL-17A level was positively correlated with FSF, CBC, and Vol. In DO-DU patients, urine 8-OHdG, PGE_2_, IL-6, IL-10, and MIP-1α levels were positively correlated with Qmax, while urine IL-5, IL-10, and MIP-1α were negatively correlated with FSF.

## 4. Discussion

To the best of our knowledge, this study is the first study investigating the most inflammatory and oxidative stress biomarkers in urine specimens of both DU and DO-DU patients. We found that both DU and DO-DU patients had different urine inflammatory and oxidative stress biomarker profiles, which might reflect the different diseased statuses inside the bladder. Not only were distinct urine biomarker profiles observed, but these biomarker levels were also found to be correlated with the urodynamic parameters among DU patients and DO-DU patients. This suggests that these urine biomarkers have the potential to map the clinical characteristics of these respective diseases. Urine inflammatory and oxidative stress biomarker analysis provides a non-invasive and convenient approach to important clinical diagnostic information in DU and DO-DU patients.

Recent research indicates that bladder and pelvic ischemia, as well as oxidative stress, might be associated with LUTS [17,18] and the development of lower urinary tract dysfunction [7,19], including DO and DU. Progression from DO to DO-DU, and ultimately DU, was proposed [8]. Therefore, oxidative stress-related biomarkers might be targets of interest in the investigation of biomarkers in both DO-DU and DU. However, it is undetermined if oxidative stress is a cause or consequence of LUTS and lower urinary tract dysfunction [20]. A study of rabbits showed that moderate bladder ischemia caused DO, whereas severe bladder ischemia caused impaired bladder contractility [21]. In the model of bladder BOO, chronic cyclic ischemia–reperfusion injury results in excessive oxidative stress and hypoxia-related inflammation, which play critical roles in disease progression and associated bladder dysfunctions, including DU [22,23]. Many animal studies have disclosed the application of oxidative stress biomarkers in BOO [23]. These analytes are potential biomarkers in BOO-related DU [9]. Hypoxia induces nuclear factor-kappa B activation and the production of reactive oxygen species, which play important roles in the etiology and progression of benign prostatic hyperplasia [24]. In a human prostate cell model, a Serenoa repens and Urtica dioica fixed combination was able to induce the decrease of the nuclear factor-kappa B inflammatory pathway and the production of inflammatory cytokines, including IL-6 and IL-8 [25]. It suggested the roles of an anti-inflammatory pathway in treating benign prostatic hyperplasia.

Oxidative stress refers to imbalanced intracellular levels of reactive oxygen species that damage lipids, proteins, and DNA. Reactive oxygen species are required for the release of pro-inflammatory cytokines, including TNFα, IL-1β, IL-6, IL-8, and IFNγ, that are involved in an appropriate immune response [26]. Eight-OHdG, a stable end-product of DNA oxidation, is a commonly used indicator of oxidative stress [17,23]. In this study, DU patients had significantly higher urine 8-OHdG, TNFα, IL-1β, and IL-8 levels than controls. Additionally, these urine biomarkers were similar between neurogenic and non-neurogenic DU patients, suggesting increased oxidative stress and oxidative stress-related inflammation in DU patients, regardless of the etiologies. In DU patients, urine TAC levels non-significantly decreased and were positively correlated with Pdet, while urine IFNγ levels were positively correlated with Pdet and negatively correlated with Vol and Qmax. TAC represents the endogenous antioxidant capacity, and urine TAC levels might be an indicator of preserved detrusor contractility in DU patients. Urine oxidative stress and its related inflammatory biomarkers might reflect intrinsic bladder functions in DU patients.

Elevated urine IL-10 levels were noted in medically refractory DO patients with the potential to differentiate DO and interstitial cystitis [15]. In this study, DO-DU patients had significantly higher urine IL-1β, IL-5, IL-6, IL-8, IL-10, IL-17A, and TNFα levels than controls. Urine 8-OHdG levels, IL-6, IL-10, and MIP-1α, were positively correlated with Qmax, while urine IL-5, IL-10, and MIP-1α levels were negatively correlated with FSF. Expressions of these biomarkers in urine might reflect increased oxidative stress and inflammation in the urinary bladder of DO-DU patients. Although there were differences in the urine oxidative stress and inflammatory cytokine profiles between DU and DO-DU patients, the hypothesis of disease progression from DO-DU to DU [8] could not be proven in this study.

PGE_2_, synthesized by cyclooxygenase-1 and cyclooxygenase-2 and released in the urinary bladder during detrusor contraction, is believed to be responsible for spontaneous detrusor contractions. It is also believed to be responsible for inducing increased excitability of afferent nerve fibers in inflamed bladder diseases states, such as neurogenic bladder, BOO, idiopathic DO, and aging bladder [27,28]. In a study investigating 75 OAB male patients, urine PGE_2_ levels were increased compared to controls and negatively correlated with maximum bladder capacity [29]. However, the OAB patients with concurrent DU (n = 7) had lower urine PGE_2_ levels than those without concurrent DU. In this study, both DU and DO-DU patients had significantly higher urine PGE_2_ levels than controls, and there was no significant difference between these two study groups. Among DU patients, urine PGE_2_ levels were positively correlated with Pdet, while within DO-DU patients, urine PGE_2_ levels were positively correlated with Qmax. Elevated urine PGE_2_ levels in both DU and DO-DU patients might reflect their respective inflammatory bladder statuses. The numerical levels of PGE_2_ might reflect residual functions involved in the signaling of detrusor contractions. Urine PGE_2_ levels might have diagnostic roles in both DU and DO-DU patients.

There were several limitations in this study. First, in order to be consistent with previous studies [14,15], enrolled controls, which were patients with genuine stress urinary incontinence, were not a perfectly healthy population, although there was no significant voiding or storage dysfunction detected in VUDS. Second, the number of DO-DU patients was small (n = 18), and there might be differences between genders within study groups. Although it is possible that gender could play a role, the impact of gender on the detected levels of inflammatory and oxidative biomarkers in urine has not been definitively proven currently. Third, there might have been intra-individual variations, including systemic factors and the different bladder conditions, when urine was collected. The etiologies of DU and DHIC are multifactorial with interplay. Systemic comorbidities might affect not only the development of DU but also the expressions of urine biomarkers. A more detailed analysis of the comorbidities is needed in the future. Fourth, the collection of fresh urine samples by self-voiding or single urethral catheterization might affect the expressions of urine biomarkers, although the infection-free status was ensured before storage. Finally, the expressions of biomarker levels in urine tended to have extreme values; therefore, we excluded them from analysis according to the previously reported operational procedures [14,15].

## 5. Conclusions

The oxidative stress biomarkers and inflammatory cytokine profiles of DU and DO-DU patients were different from the controls. These urine analyte levels also correlated with the urodynamic parameters in both DU and DO-DU patients. Urine analytes associated with inflammation and oxidative stress, with the potential to be biomarkers of DU and DO-DU, might provide important intrinsic bladder information in these patients.

## Figures and Tables

**Figure 1 biomedicines-11-01191-f001:**
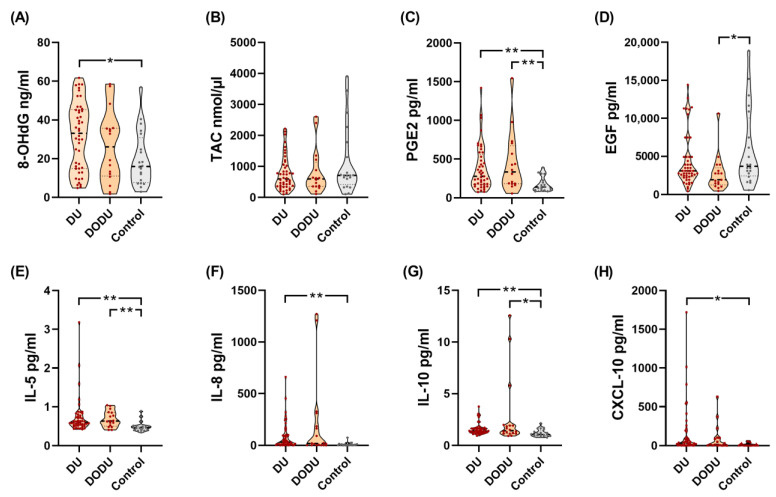
Violin plots of significant analytes discriminating DU, DO-DU patients, and controls. DU, detrusor underactivity. (**A**) Urine 8-OHdG levels, (**B**) Urine TAC levels, (**C**) Uriine PGE2 levels, (**D**) Urine EGF levels, (**E**) Urine IL-5 levels, (**F**) Urine IL-8 levels, (**G**) Urine IL-10 levels, (**H**) Urine CXCL10 levels; DO-DU, detrusor overactivity and detrusor underactivity; 8-OHdG, 8-hydroxy-2-deoxyguanosine; TAC, total antioxidant capacity; PGE2, prostaglandin E2; EGF, epidermal growth factor; CXCL10, chemokine (C-X-C motif) ligand 10. * *p* < 0.05, ** *p* < 0.01.

**Table 1 biomedicines-11-01191-t001:** Clinical characteristics of DU, DO-DU patients, and controls.

	(A) DU (n = 50)	(B) DO-DU (n = 18)	(C) Controls (n = 20)	*p* Value	Post-Hoc Analysis
Age	63.4 ± 14.0	65.3 ± 13.7	64.4 ± 8.5	0.628	
Sex	F 33, M 17	F 9, M 9	F 20	0.002	A, B vs. C
IPSS-V	15.2 ± 4.2	12.5 ± 8.1	1.3 ± 1.5	<0.001	A, B vs. C
IPSS-S	5.9 ± 4.6	7.7 ± 4.3	1.9 ± 1.4	<0.001	A, B vs. C
IPSS	21.1 ± 6.6	20.2 ± 10.4	3.2 ± 1.5	<0.001	A, B vs. C
**VUDS**					
FSF	204.6 ± 98.4	123.3 ± 56.1	178.8 ± 76.1	0.005	A vs. B
FS	313.4 ± 108.0	196.2 ± 85.2	314.0 ± 91.1	<0.001	B vs. A, C
CBC	445.6 ± 245.9	374.2 ± 119.3	419.7 ± 146.7	0.206	
Pdet	3.9 ± 7.7	20.0 ± 13.9	14.0 ± 7.2	<0.001	A vs. B, C
Qmax	4.7 ± 5.3	5.9 ± 3.9	18.9 ± 5.3	<0.001	A, B vs. C
cQmax	0.23 ± 0.28	0.30 ± 0.21	0.9 ± 0.3	<0.001	A, B vs. C
Vol	101.6 ± 137.4	130.0 ± 103.2	436.0 ± 112.1	<0.001	A, B vs. C
PVR	362.0 ± 258.5	244.2 ± 162.1	5.8 ± 13	<0.001	A, B vs. C
VE	0.25 ± 0.34	0.38 ± 0.30	1.0 ± 0.1	<0.001	A, B vs. C

DU, detrusor underactivity; DO-DU, detrusor overactivity and detrusor underactivity; IPSS, International Prostate Symptom Score; IPSS-S, IPSS storage subscore; IPSS-V, IPSS voiding subscore; VUDS, videourodynamic study; FSF, first sensation of bladder filling; FS, fullness sensation; CBC, cystometric bladder capacity; Pdet, detrusor voiding pressure; Qmax, maximal urinary flow rate; cQmax, corrected maximal urinary flow rate; Vol, voided volume; PVR, post-void residual volume; VE, voiding efficacy.

**Table 2 biomedicines-11-01191-t002:** Urine biomarker levels among DO, DO-DU patients, and controls.

Urine Cytokines ^@^	(A) DU (n = 50)	(B) DO-DU (n = 18)	(C) Control (n = 20)	*p* Value	Post-Hoc Analysis
8-OHdG	31.67 ± 17.52 (4)	26.28 ± 18.6 (2)	18.19 ± 14.65 (0)	0.018	A vs. C
8-isoprostane	23.80 ± 16.39 (4)	20.14 ± 9.30 (3)	20.23 ± 21.65 (0)	0.290	
TAC	765.2 ± 547.5 (4)	818.6 ± 739.6 (2)	1427.4 ± 1202.3 (0)	0.156	
PGE_2_	384.3 ± 295.8 (5)	449.3 ± 388.7 (2)	166.78 ± 75.71 (1)	0.001	A, B vs. C
EGF	4334.8 ± 3445.0 (0)	2740.2 ± 2382.4 (1)	6967.3 ± 4749.9 (0)	0.002	B vs. C
Eotaxin	8.25 ± 9.88 (1)	5.82 ± 7.64 (0)	5.23 ± 4.41 (0)	0.551	
G-CSF	29.84 ± 58.95 (1)	110.8 ± 200.51 (0)	9.00 ± 6.47 (0)	0.195	
GM-CSF *	1.50 ± 0.70 (1)	1.48 ± 0.64 (1)	1.20 ± 0.38 (1)	0.140	
IFNα2	3.46 ± 1.71 (2)	2.88 ± 0.98 (1)	3.24 ± 1.48 (0)	0.631	
IFNγ	1.18 ± 0.4 (1)	1.11 ± 0.36 (1)	1.17 ± 0.19 (0)	0.501	
IL-1RA	416.2 ± 492.6 (1)	333.2 ± 421.0 (1)	229.3 ± 300.6 (1)	0.225	
IL-1α *	2.13 ± 1.93 (1)	2.53 ± 3.00 (1)	1.47 ± 0.77 (0)	0.389	
IL-1β	1.57 ± 2.48 (1)	33.74 ± 118.08 (1)	0.47 ± 0.14 (1)	0.006	A, B vs. C
IL-2 *	0.74 ± 0.18 (1)	0.73 ± 0.24 (1)	0.80 ± 0.16 (0)	0.217	
IL-3 *	0.44 ± 0.26 (0)	0.35 ± 0.22 (0)	0.63 ± 0.21 (0)	0.001	A, B vs. C
IL-4	11.94 ± 13.60 (2)	7.55 ± 5.95 (0)	12.82 ± 17.90 (1)	0.326	
IL-5	0.73 ± 0.47 (2)	0.68 ± 0.21 (1)	0.51 ± 0.19 (0)	0.001	A, B vs. C
IL-6	3.19 ± 6.00 (2)	5.60 ± 9.14 (1)	0.84 ± 0.31 (1)	0.023	B vs. C
IL-7	1.63 ± 0.74 (2)	1.54± 0.70 (0)	1.34 ± 0.60 (1)	0.164	
IL-8	84.28 ± 129.28 (1)	202.47 ± 392.22 (0)	12.91 ± 22.05 (0)	0.003	A, B vs. C
IL-10	1.6 ± 0.58 (1)	2.77 ± 3.36 (0)	1.22 ± 0.34 (0)	0.005	A, B vs. C
IL-12p40 *	1.13 ± 0.66 (0)	1.6 ± 1.82 (0)	0.72 ± 0.32 (0)	0.055	
IL-12P70	1.29 ± 0.33 (1)	1.3 ± 0.45 (0)	1.24 ± 0.23 (0)	0.788	
IL-13 *	1.29 ± 0.64 (1)	1.11 ± 0.28 (1)	1.24 ± 0.40 (0)	0.633	
IL-15	1.81 ± 1.06 (1)	2.03 ± 1.24 (0)	1.34 ± 0.46 (0)	0.165	
IL-17A	1.61 ± 1.91 (1)	3.07 ± 3.27 (0)	0.88 ± 0.13 (0)	0.012	A, B vs. C
CXCL10	148.65 ± 311.08 (1)	95.93 ± 169.13 (1)	15.16 ± 19.91 (0)	0.027	A vs. C
MCP-1	293.23 ± 292.72 (1)	278.92 ± 187.81 (0)	168.26 ± 109.93 (0)	0.190	
MIP-1α	2.7 ± 5.85 (1)	6.93 ± 14.52 (1)	1.22 ± 0.8 (0)	0.277	
MIP-1β	5.17 ± 5.66 (1)	10.09 ± 19.4 (0)	2.86 ± 2.42 (0)	0.303	
RANTES	16.87 ± 36.05 (1)	9.39 ± 10.56 (0)	5.84 ± 3.52 (0)	0.376	
TNFα	2.18 ± 3.11 (1)	4.81 ± 8.5 (1)	0.76 ± 0.22 (0)	0.030	
TNFβ *	0.82 ± 0.15 (1)	0.77 ± 0.17 (0)	0.77 ± 0.10 (0)	0.148	
VEGF *	13.28 ± 8.45 (1)	14.36 ± 15.18 (1)	11.58 ± 5.05 (0)	0.690	
NGF *	0.25 ± 0.1 (0)	0.23 ± 0.13 (1)	0.27 ± 0.07 (0)	0.159	
BDNF	0.95 ± 0.93 (2)	0.72 ± 0.24 (1)	0.60 ± 0.16 (0)	0.283	

8-OHdG, 8-hydroxy-2-deoxyguanosine; TAC, total antioxidant capacity; PGE2, prostaglandin E2; EGF, epidermal growth factor; G-CSF, granulocyte colony-stimulating factor; GM-CSF, granulocyte–macrophage colony-stimulating factor; IFN, interferon; IL, interleukin; IL-1RA, IL-1 receptor antagonist; CXCL10, chemokine (C-X-C motif) ligand 10; MCP-1, macrophage chemoattractant protein-1; MIP, macrophage inflammatory protein; RANTES, regulated upon activation, normal T cell expressed and presumably secreted; VEGF, vascular endothelial growth factor; TNF, tumor necrosis factor; NGF, nerve growth factor; BDNF, brain-derived neurotrophic factor. (): indicates the number of outliers. *: Mean values of the study group that were below the minimum detectable concentrations as per the assay manufacturer. @: units: all pg/mL, except for ng/mL in 8-OHdG and mmol/μL in TAC.

**Table 3 biomedicines-11-01191-t003:** Multivariate models (controlling for age and sex) reveal the diagnostic values of targeted urine biomarkers in DU.

	*p* Value	Odds Ratio	95% CI	Odds Ratio Units *
**DU vs. control**				
PGE_2_	0.026	2.165	1.096–4.275	100
8-OHdG	0.023	1.557	1.062–2.282	10
IL-5	0.039	1.544	1.022–2.362	0.1
IL-10	0.015	1.318	1.056–1.644	0.1
IL-8	0.047	1.291	1.004–1.660	10
TAC	0.014	0.904	0.834–0.980	100
EGF	0.019	0.826	0.704–0.969	1000

PGE2, prostaglandin E2; 8-OHdG, 8-hydroxy-2-deoxyguanosine; TAC, total antioxidant capacity; EGF, epidermal growth factor; *: units: all pg/mL, except for ng/mL in 8-OHdG and mmol/μL in TAC.

**Table 4 biomedicines-11-01191-t004:** The correlation coefficient (r value) between urine biomarker levels and VUDS parameters in patients with DU.

Urine Cytokines	FSF	FS	CBC	PVR	Vol	Qmax	cQmax	Pdet
TAC	n.s.	n.s.	n.s.	n.s.	n.s.	n.s.	n.s.	0.350
PGE2	n.s.	n.s.	n.s.	n.s.	n.s.	n.s.	n.s.	0.391
G-CSF	n.s.	n.s.	n.s.	n.s.	0.370	n.s.	n.s.	n.s.
IFNγ	n.s.	n.s.	n.s.	n.s.	0.325	0.315	0.308	−0.315
IL-17A	0.323	n.s.	0.376	n.s.	0.338	n.s.	n.s.	n.s.
MCP-1	n.s.	n.s.	n.s.	n.s.	−0.352	−0.391	−0.363	n.s.

VUDS, videourodynamic study; DU, detrusor underactivity; FSF, first sensation of bladder filling; FS, fullness sensation; CBC, cystometric bladder capacity; Pdet, detrusor voiding pressure; Qmax, maximal urinary flow rate; cQmax, corrected maximal urinary flow rate; Vol, voided volume; PVR, post-void residual volume; TAC, total antioxidant capacity; PGE2, prostaglandin E2; G-CSF, granulocyte colony-stimulating factor; IFNγ, interferon γ; MCP-1, macrophage chemoattractant protein-1; n.s., not significant.

**Table 5 biomedicines-11-01191-t005:** The correlation coefficient (r value) between urine biomarker levels and VUDS parameters in patients with DO-DU.

Urine Cytokines	FSF	FS	CBC	PVR	Vol	Qmax	cQmax	Pdet
8-OHdG	n.s.	n.s.	n.s.	n.s.	n.s.	0.542	0.567	n.s.
PGE2	n.s.	n.s.	n.s.	n.s.	n.s.	0.644	0.602	n.s.
IFNγ	0.596	n.s.	n.s.	n.s.	n.s.	n.s.	n.s.	n.s.
IL-5	−0.541	n.s.	n.s.	n.s.	n.s.	n.s.	n.s.	n.s.
IL-6	n.s.	n.s.	n.s.	n.s.	n.s.	0.494	n.s.	n.s.
IL-7	n.s.	n.s.	n.s.	0.541	n.s.	n.s.	n.s.	n.s.
IL-8	n.s.	n.s.	0.518	n.s.	n.s.	n.s.	n.s.	n.s.
IL-10	−0.555	n.s.	n.s.	n.s.	n.s.	0.646	0.583	n.s.
IL-12p70	n.s.	n.s.	n.s.	0.480	n.s.	n.s.	n.s.	n.s.
IL-15	n.s.	−0.517	0.597	0.533	n.s.	n.s.	n.s.	n.s.
MIP-1α	−0.581	n.s.	n.s.	n.s.	n.s.	0.649	0.591	n.s.
BDNF	n.s.	0.538	n.s.	n.s.	n.s.	n.s.	n.s.	n.s.

VUDS, videourodynamic study; DU, detrusor underactivity; FSF, first sensation of bladder filling; FS, fullness sensation; CBC, cystometric bladder capacity; Pdet, detrusor voiding pressure; Qmax, maximal urinary flow rate; cQmax, corrected maximal urinary flow rate; Vol, voided volume; PVR, post-void residual volume; 8-OHdG, 8-hydroxy-2-deoxyguanosine; PGE2, prostaglandin E2; IFNγ, interferon γ; MIP-1α, macrophage inflammatory protein 1α; BDNF, brain-derived neurotrophic factor; n.s., not significant.

## Data Availability

Data sharing is not applicable to this article due to ethical restrictions.

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
