# Peer review of "Investigating Urine Biomarkers in Detrusor Underactivity and Detrusor Overactivity with Detrusor Underactivity Patients"

_biomedicines, 2023, doi:10.3390/biomedicines11041191_

Round 1
Reviewer 1 Report
This paper is well written, interesting and original. It is the first study investigating inflammatory and oxidative stress biomarkers in the urine of patients with detrusor overactivity and detrusor underactivity patients. These biomarkers were found different from controls (unfortunately not healthy patients, but patients with stress incontinence) and correlated with the urodynamic parameters of the patients
Even if this study did not focus on the clinical consequences of this discovery, the clinical implications of this research are clear and will hopefully be the topic of a future paper of the Authors.
For example, It would be interesting to see if there are changes in the biomarkers of patients with benign prostatic hyperplasia (BPH), looking for the biomarkers highlighted by the Authors, from the initial stages of the disease, when the detrusor contracts well and there might be an overacting detrusor as a consequence of obstruction, to the final stages, when the detrusor becomes underactive and the PVR of the patients becomes gradually higher. Detrusor underactivity is a strongly related to evolution of BPH. Moreover, the suggested biomarkers could be used to monitor the efficacy of drugs able to relieve BPH symptoms, too see of they are really active and able to protect the detrusor activity, or, on the other hand, if there is a deterioration of the detrusor activity (detected by the biomarkers), so that the drug regimen can be updated, increased or stopped definitively, sending the patient for endoscopic prostate resection (TURP). The Authors should comment briefly in the discussion on this important clinical point. BPH etiology and progression have indeed been associated with the persistence of an inflammatory stimulus induced by NF-KB activation and ROS production. Inhibition of these pathways by specific drugs has been shown to be beneficial to patients with BPH. In a recent publication on IJMS, anti-oxidant and anti-inflammatory activity of these drugs has been shown in an in vitro human model of BPH (Mancini M., et al, Int J Mol Sci. 2020, 21, 9178; doi:103390/ijms211239178). Correlation of deterioration of detrusor activity and prostate inflammatory status in BPH, and the use of urine specific biomarkers to detects these changes, seems to be an innovative and promising line of research.
Author Response
Reviewer 1
This paper is well written, interesting and original. It is the first study investigating inflammatory and oxidative stress biomarkers in the urine of patients with detrusor overactivity and detrusor underactivity patients. These biomarkers were found different from controls (unfortunately not healthy patients, but patients with stress incontinence) and correlated with the urodynamic parameters of the patients.
Reply: Thank you for the comment.
In previously studies about urine biomarker studies (ref. 14, 15, 16), they designed to enroll the female SUI patients (with only urethral competence, but without other significant storage or voiding dysfunction in VUDS) as controls. In order to maintain the consistency of the inclusion criteria of controls, we set the similar inclusion criteria of controls. We have mentioned the limitations in Discussion.
Even if this study did not focus on the clinical consequences of this discovery, the clinical implications of this research are clear and will hopefully be the topic of a future paper of the Authors.
For example, It would be interesting to see if there are changes in the biomarkers of patients with benign prostatic hyperplasia (BPH), looking for the biomarkers highlighted by the Authors, from the initial stages of the disease, when the detrusor contracts well and there might be an overacting detrusor as a consequence of obstruction, to the final stages, when the detrusor becomes underactive and the PVR of the patients becomes gradually higher. Detrusor underactivity is a strongly related to evolution of BPH. Moreover, the suggested biomarkers could be used to monitor the efficacy of drugs able to relieve BPH symptoms, too see of they are really active and able to protect the detrusor activity, or, on the other hand, if there is a deterioration of the detrusor activity (detected by the biomarkers), so that the drug regimen can be updated, increased or stopped definitively, sending the patient for endoscopic prostate resection (TURP). The Authors should comment briefly in the discussion on this important clinical point. BPH etiology and progression have indeed been associated with the persistence of an inflammatory stimulus induced by NF-KB activation and ROS production. Inhibition of these pathways by specific drugs has been shown to be beneficial to patients with BPH. In a recent publication on IJMS, anti-oxidant and anti-inflammatory activity of these drugs has been shown in an in vitro human model of BPH (Mancini M., et al, Int J Mol Sci. 2020, 21, 9178; doi:103390/ijms211239178). Correlation of deterioration of detrusor activity and prostate inflammatory status in BPH, and the use of urine specific biomarkers to detects these changes, seems to be an innovative and promising line of research.
Reply: Thank you for the comment.
We have added the related descriptions and cited the references in the 2nd paragraph of “Discussion” (Hypoxia induces nuclear factor-kappa B activation and the production of reactive ox-ygen species, which play important roles in the etiology and progression of benign prostatic hyperplasia (ref. 24). In a human prostate cell model, Serenoa repens and Urtica dioica fixed combination was able to induce the decrease of the nuclear fac-tor-kappa B inflammatory pathway and the production of inflammatory cytokines, in-cluding IL-6 and IL-8 (ref. 25). It suggested the roles of anti- inflammatory pathway in treating benign prostatic hyperplasia.)
Reviewer 2 Report
Interesting study, with results being part of a larger prospective study on urine biomarkers that the authors have conducted. Inclusion and exclusion criteria are well described. Results could be of clinical importance, pending some methodological issues, which the authors mostly acknowledge in the limitations section. Other than that, this is a well written manuscript, in a very informative and comprehensive fashion and good Introduction and Discussion sections.
Main issues to address:
1. 1..Unequal group sizes for a prospective study. Significantly more patients were recruited in the DU group than in the DO+DU and controls groups. The authors already acknowledge this as a limitation to their study, in addition to gender and age matching issues.
2. 2. Since this was a prospective study, primary and secondary outcomes would be needed. The authors could use preliminary results or published literature to identify a key biomarker upon which to also base sample size calculation, including what is thought to be significant difference between study groups and controls. Please include in Methods.
Author Response
Reviewer 2
Interesting study, with results being part of a larger prospective study on urine biomarkers that the authors have conducted. Inclusion and exclusion criteria are well described. Results could be of clinical importance, pending some methodological issues, which the authors mostly acknowledge in the limitations section. Other than that, this is a well written manuscript, in a very informative and comprehensive fashion and good Introduction and Discussion sections.
Main issues to address:
- Unequal group sizes for a prospective study. Significantly more patients were recruited in the DU group than in the DO+DU and controls groups. The authors already acknowledge this as a limitation to their study, in addition to gender and age matching issues.
Reply: Thank you for the comment.
We have mentioned these limitations in the last paragraph of “Discussion”. (Second, the number of DO-DU patients were small (n = 18) and there might be differences between genders within study groups.)
- Since this was a prospective study, primary and secondary outcomes would be needed. The authors could use preliminary results or published literature to identify a key biomarker upon which to also base sample size calculation, including what is thought to be significant difference between study groups and controls. Please include in Methods.
Reply: Thank you for the comment.
We have performed post-hoc power calculation in the biomarkers with significant difference between the study and control groups (mentioned in Materials and Methods section, the paragraph of 2.3 Statistical analysis). (Post-hoc power calculation was performed in the biomarker with significant difference between the study and control groups.)
We also added the calculated post-hoc power in the results (2nd paragraph of Results section). (Post-hoc power analysis reported 90.7%, 99.8%, 61.5%, 89.4%, 87.8%, 79.6%, 78.8%, 96.5%, 92.5%, 76.6%, and 85.5% power (with alpha value of 0.05) in the evaluation of 8-OHdG, PGE2, EGF, TNFα, IL-1β, IL-5, IL-6, IL-8, IL-10, IL-17A, and CXCL-10 levels in DU patients, respectively.)
Reviewer 3 Report
The study is well conceived, but there are some issues to be resolved. In table 1 gender has a significant statistical significance, with no male in the control group. The whole premise of the paper depends on this. I would like to see the explanation from the authors, because this was not commented on in the results, but was later used in the analysis.
Can the authors explain why they chose the ANOVA test? Are we sure that all groups have a normal distribution and no difference in covariance between groups? Some of the values in Table have huge differences between three groups, namely the Vol (voided volume) and PVR (post-void residual volume)? Please comment on this data in Table 1.
The authors show a correlation matrix in table 4 and table 5 with r values. What is the statistical test used for this correlation?
Author Response
Reviewer 3
The study is well conceived, but there are some issues to be resolved. In table 1 gender has a significant statistical significance, with no male in the control group. The whole premise of the paper depends on this. I would like to see the explanation from the authors, because this was not commented on in the results, but was later used in the analysis.
Reply: Thank you for the comment.
The absence of clinical symptoms does not necessarily indicate the absence of a lower urinary tract disease, which is needed to be confirmed by the gold standard study- VUDS. But, it is difficult and not feasible to perform VUDS in the patients without significant LUTS. Therefore, in previously studies about urine biomarker studies (ref. 14, 15, 16), they designed to enroll the female SUI patients (with only urethral competence, but without other significant storage or voiding dysfunction in VUDS) as controls. In order to maintain the consistency of the inclusion criteria of controls, we set the similar inclusion criteria of controls. We have added the description “Significant difference in gender was noted among groups.” in the 1st paragraph of “Results”.
Can the authors explain why they chose the ANOVA test? Are we sure that all groups have a normal distribution and no difference in covariance between groups? Some of the values in Table have huge differences between three groups, namely the Vol (voided volume) and PVR (post-void residual volume)? Please comment on this data in Table 1.
Reply: Thank you for the comment.
- The expressions of most targeted analytes had a normal distribution (including 8-OHdG, GM-CSF, IL-10, IL-12p40, IL-13, IL-5, MIP-1α, and VEGF in study group, and 8-OHdG, GM-CSF, IFNα2, IFNγ, IL-12p70, IL-13, IL-15, IL-17A, IL-1α, IL-1β, IL-2, IL-3, IL-4, IL-7, MCP-1, MIP-1α, RANTES, TNF-α, TNFβ, VEGF, NGF, and BDNF in control group). In addition, we excluded the outliers for further analysis as previous studies (ref. 14, 15, 16). It seemed to us that current statistic methods were available.
- As we know, the majority of DU patients might have “decreased” bladder sensation, and DO-DU patients had DU and DO with presumed “increased” bladder sensation. Under the conditions of low detrusor contractility (DU) with low voided efficacy (i.e. Vol./ (Vol. + PVR)), the SD of Vol. and PVR within study groups would be large. And the huge differences of the SD of Vol. and PVR among these groups reflects their distinct detrusor contractility and sensation functions, and it is reasonable.
The authors show a correlation matrix in table 4 and table 5 with r values. What is the statistical test used for this correlation?
Reply: Thank you for the comment.
We have added the description in Materials and Methods (the paragraph of 2.3 Statistical analysis). (To assess the correlation between clinical characteristics and urine biomarker levels, a linear regression analysis using Pearson correlation was conducted.)
Reviewer 4 Report
The authors prepared an interesting clinical study on DU/DO-DU individuals that were characterized by biomarkers analyzed in their urine. The analyzed cohort is rather small and heterogenous. Furthermore, several doubts and comments were listed below, especially concerning the methodology.
Please revise the title to be more informative and specific and correct it at the same time, e.g. the urine markers of detrusor underactivity (and overactivity) in underactive bladder patients.
Please make a clear statement concerning the focus on male patients in section 2.
Can you provide the reference for your definition of DU? It is not based on the validated VUDS calculations nor nomograms and it is not gender specific.
Why did you use females with SUI as controls? There can be great differences, especially concerning the urine biomarkers.
Concerning the exclusion criteria, why did not you exclude CIC patients?
As for the DU and DO-DU groups, were there any comorbidities or you expected the VUDS findings as idiopathic as you excluded BOO in cystoscopies? Did you find any neurogenic bladders? As you stated that you found that both DU and DO-DU patients had different urine inflammatory and oxidative stress biomarker profiles, which might reflect the different diseased statuses inside the bladder, it is worth adding additional clinical data to those of urodynamic origin. Also commenting that in the discussion is necessary.
Please further comment on which grounds you excluded the expressions of biomarker levels in urine from analysis which tended to have extreme values according to the previously reported operational procedures.
As for the collection of urine, in which individuals did you collect fresh voided sample and in which you needed catheterization? Were there any differences concerning the biomarkers?
Author Response
Reviewer 4
The authors prepared an interesting clinical study on DU/DO-DU individuals that were characterized by biomarkers analyzed in their urine. The analyzed cohort is rather small and heterogenous. Furthermore, several doubts and comments were listed below, especially concerning the methodology.
Please revise the title to be more informative and specific and correct it at the same time, e.g. the urine markers of detrusor underactivity (and overactivity) in underactive bladder patients.
Reply: Thank you for the comment.
Underactive bladder is clinical diagnosis based on clinical symptoms. Both DU and DO-DU are the diagnosis based on urodynamic studies. In this article, we enrolled the patients based on VUDS (urodynamic studies). Therefore, to keep current title “Investigating urine biomarkers in DU and DODU patients” (under the diagnosis of VUDS) is more feasible.
Please make a clear statement concerning the focus on male patients in section 2.
Reply: Thank you for the comment. We have added the description in the 1st paragraph of “Results.” (Significant difference in gender was noted among groups.) Please let us know more clearly, about the necessity of additional statement (in Methods/ Results/ or Discussion?, and which related issue?)
Can you provide the reference for your definition of DU? It is not based on the validated VUDS calculations nor nomograms and it is not gender specific.
Reply: Thank you for the comment.
Currently, there is no consensus about the clear operational definition of DU (ICS definition, ref. 2 Neurourol Urodyn 2019, 38 (2), 433-477.). Some study used the definition of bladder contractility index < 100; however, it is not feasible. In this study, we set “stricter” definitions in order to enroll “more typical” DU patients (low detrusor pressure, low urinary flow, and large PVR), and this would help us successfully to explore the roles of biomarkers in these patients.
In “2.1 Patients and investigation of the clinical characteristics”, we have mentioned that “In this study, we set stricter conditions to enroll more typical study patients.”
Why did you use females with SUI as controls? There can be great differences, especially concerning the urine biomarkers.
Reply: Thank you for the comment.
The absence of clinical symptoms does not necessarily indicate the absence of a lower urinary tract disease, which is confirmed by the gold standard VUDS. However, it is difficult and not feasible to perform VUDS in the patients without significant LUTS. Therefore, in previously studies about urine biomarker studies (ref. 14, 15, 16), they designed to enroll the female SUI patients (with only urethral competence, but without other significant storage or voiding dysfunction in VUDS) as controls. In order to maintain the consistency of the inclusion criteria of controls, we set the similar inclusion criteria of controls. We have mentioned the limitations in Discussion.
Concerning the exclusion criteria, why did not you exclude CIC patients?
Reply: Thank you for the comment. In this study, we set stricter definitions in order to enroll more typical DU patients (low detrusor pressure, low urinary flow, and large PVR), and this would help us more successfully to explore the biomarkers in these patients. Inevitably, we would enroll some DU patients needing CIC. Therefore, we conducted urinalysis to ensure an infection-free status before storing the urine samples, to reduce the impact from CIC. In the section of ”2.2. Assessment of urine biomarker levels”, we have mentioned that “To ensure an infection-free status, urinalysis was conducted before storing the fresh urine samples.” Additionally, as reported in ref. 12, some urine samples were collected were collected for biomarker investigation by CIC.
As for the DU and DO-DU groups, were there any comorbidities or you expected the VUDS findings as idiopathic as you excluded BOO in cystoscopies? Did you find any neurogenic bladders? As you stated that you found that both DU and DO-DU patients had different urine inflammatory and oxidative stress biomarker profiles, which might reflect the different diseased statuses inside the bladder, it is worth adding additional clinical data to those of urodynamic origin. Also commenting that in the discussion is necessary.
Reply: Thank you for the comment.
- The etiologies of DU and DHIC are multifactorial with interplay. We enrolled the patients based on VUDS findings and performed cystoscopy to exclude BOO (already mentioned in Methods section). We could not clearly determine the etiologies of DU by VUDS or cystoscopy.
Systemic comorbidities might affect not only the development of DU but also the expressions of urine biomarkers. We did not analyze the systemic factors, and we have added this into the paragraph of limitation. (“The etiologies of DU and DHIC are multifactorial with interplay. Systemic comorbidities might affect not only the development of DU but also the expressions of urine biomarkers. More detailed analysis about the comorbidities is needed in the future.”)
- We have mentioned that “There was no significant difference in urine biomarker profiles between DU patients with different etiologies (neurogenic or non-neurogenic).” In the 2nd paragraph of Results.
- We have added the more statement in the 1st paragraph of Discussion. (Not only were distinct urine biomarker profiles observed, but these biomarker levels were also found to be correlated with the urodynamic parameters among DU patients and DO-DU patients. This suggests that these urine biomarkers have the potential to map the clinical characteristics of these respective diseases.)
Please further comment on which grounds you excluded the expressions of biomarker levels in urine from analysis which tended to have extreme values according to the previously reported operational procedures.
Reply: Thank you for the comment.
We have provided the related citations (ref. 14, 15, previous studies using the similar operational procedures to exclude extreme values) in 2.3 statistical analysis.
As for the collection of urine, in which individuals did you collect fresh voided sample and in which you needed catheterization? Were there any differences concerning the biomarkers?
Reply: Thank you for the comment.
- We collect “fresh” urine by self voiding or CIC, and the fresh urine samples were transported to laboratory for processing. We have added the note of ”fresh” urine samples in the section of ”2.2. Assessment of urine biomarker levels”.
- First, we conducted urinalysis to ensure an infection-free status before storing the urine samples, to reduce the impact from CIC. Second, the collection of fresh urine samples by self voiding or single urethral catheterization might affect the expressions of urine biomarkers. Nonetheless, as reported in ref. 12, some urine samples were also collected for biomarker investigation by CIC.
We have added this limitation in the last paragraph of Discussion. (“Fourth, the collection of fresh urine samples by self voiding or single urethral catheterization might affect the expressions of urine biomarkers, although the infection-free status was ensured before storage.”)
Round 2
Reviewer 3 Report
Dear authors,
Thank you for your kind revisions, the manuscript is much improved. Can you please comment on the gender differences between groups enrolled in the study in a short comment?
Why was the Pearson correlation chosen and not Spearman? Please highlight the data in order of significance (strong correlation, moderate, or low correlation along the R factor).
Author Response
Reviewer 3
Dear authors,
Thank you for your kind revisions, the manuscript is much improved. Can you please comment on the gender differences between groups enrolled in the study in a short comment?
Reply: Thank you for the comment.
We have added the comments in the last paragraph of Discussion (limitations).
“…there might be differences between genders within study groups. Although it is possible that gender could play a role, the impact of gender on the detected levels of inflammatory and oxidative biomarkers in urine has not been definitively proven currently.”
Why was the Pearson correlation chosen and not Spearman? Please highlight the data in order of significance (strong correlation, moderate, or low correlation along the R factor).
Reply: Thank you for the comments.
- We chose Pearson correlation for several reasons:
- The expressions of these biomarkers in urine were continuous variables.
- Some biomarkers were proven to be normally distributed.
- Spearman correlation is more feasible in the analysis with outliers/ extreme values. We have excluded the outliers for further analysis.
- We have added the description of the significance of correlation in the text (4th paragraph of Results “In DU patients, the significant correlations were weak; however, in DO-DU patients, the significant correlations were moderate to strong.”) not in the table (to avoid difficult reading for readers).
Reviewer 4 Report
The authors improved the paper sufficiently and responded to all the queries.
Author Response
Reviewer 4
The authors improved the paper sufficiently and responded to all the queries.
Reply: Thank you for the comment.